# Metabolomic Study of a Rat Model of Retinal Detachment

**DOI:** 10.3390/metabo12111077

**Published:** 2022-11-07

**Authors:** Xiangjun She, Yifan Zhou, Zhi Liang, Jin Wei, Bintao Xie, Yun Zhang, Lijun Shen

**Affiliations:** 1School of Ophthalmology, Optometry and Eye Hospital, Wenzhou Medical University, Wenzhou 325027, China; 2Department of Ophthalmology, Putuo Poeple’s Hospital, Tongji University, Shanghai 200070, China; 3Department of Ophthalmology, Shanghai General Hospital, National Clinical Research Center for Eye Diseases, Shanghai 200080, China; 4State Key Laboratory of Optometry, Ophthalmology, and Vision Science, Wenzhou 325027, China; 5Zhejiang Provincial People’s Hospital, Affiliated People’s Hospital of Hangzhou Medical College, Shangtang Road 158#, Gongshu District, Hangzhou 310014, China

**Keywords:** retinal detachment, metabolomics, amino acid, histamine, retina degeneration

## Abstract

Retinal detachment is a serious ocular disease leading to photoreceptor degeneration and vision loss. However, the mechanism of photoreceptor degeneration remains unclear. The aim of this study was to investigate the altered metabolism pathway and physiological changes after retinal detachment. Eight-week-old male SD rats were fed, and the model of retinal detachment was established by injecting hyaluronic acid into the retinal space. The rats were euthanized 3 days after RD, and the retinal tissues were sectioned for analysis. Untargeted lipid chromatography-mass spectrometry lipidomic was performed to analyze the metabolite changes. A total of 90 significant metabolites (34 in anionic and 56 in cationic models) were detected after retinal detachment. The main pathways were (1) histidine metabolism; (2) phenylalanine, tyrosine, and tryptophan biosynthesis; and (3) glycine, serine, and threonine metabolism. The key genes corresponding to each metabolic pathway were verified from the Gene Expression Omnibus (GEO) database of human retinal samples. The results indicated that the production of histamine by histidine decarboxylase from histidine reduced after RD (*p* < 0.05). Xanthine, hypoxanthine, guanine, and guanosine decreased after RD (*p* < 0.05). Decreased xanthine and hypoxanthine may reduce the antioxidant ability. The decreased guanosine could not provide enough sources for inosine monophosphate production. Tyrosine is an important neurotransmitter and was significantly reduced after RD (*p* < 0.05). Citrate was significantly reduced with the increase of ATP-citrate lyase enzyme (ACLY) (*p* < 0.05). We inferred that lipid oxidation might increase rather than lipid biogenesis. Thus, this study highlighted the main changes of metabolite and physiological process after RD. The results may provide important information for photoreceptor degeneration.

## 1. Introduction

Retinal detachment results from the separation of the retinal neurosensory layer from the retinal pigment epithelium layer and is commonly detected in rhegmatogenous retinal detachment, diabetic retinopathy, age-related macular degeneration, and other retinal disorders [1,2]. With the rapid development of surgical intervention for rhegmatogenous retinal detachment, higher rates of successful surgery could be achieved for most patients. However, over 50% of the patients could not achieve good visual acuity even after successful reattachment surgery [3].

Photoreceptor degeneration leading to cell death has been considered as the major reason for progressive visual impairment after retinal detachment [4]. The retinal pigment epithelium (RPE) infiltrates into the vitreous and begins to join in the process of proliferative vitreoretinopathy (PVR), which can destroy retinal integrity and cause photoreceptor degeneration [5]. However, the underlying mechanism of photoreceptor degeneration was unclear until now, and it is necessary to identify the mechanism.

Metabolomics reflects the physiological and pathological process for disease and has been widely used to identify the key networks of metabolic files in diabetic retinopathy, cancer, and other diseases [6]. Metabolomics helps in understanding the interactions between genes and proteins [6]. Previous studies analyzed the vitreous samples from RD and PVR patients and the results pointed that, after RD, the main changes were inflammation, proliferation, and energy consumption, like the significant alterations of L-carnitine, ascorbate, and valine [7,8]. Increased citric acid cycle metabolism is detected in patients with choroid and rhegmatogenous retinal detachment [9]. Another study pointed that the adenosine and inosine were increased from the vitreous in patients with retinal detachment than epiretinal membrane [10]. However, these studies only analyzed vitreous samples from patients, not retinal samples. In addition, there were no results to reveal the early metabolomic changes of RD. Humans used rodent models of acute retinal detachment to investigate the cellular events for the difficulty in obtaining human retina tissues. The rats are the ideal model for lower ethical and financial costs [11]. Our study established the rat model of RD and directly analyzed the early changes in the retina, which may reveal the key changes and help to understand the detailed mechanism for RD.

In the present study, we used untargeted metabolomics to analyze the global metabolomic profiles in a rat model of retinal detachment. It revealed 90 discriminant metabolites based on high-resolution mass spectrometry. The study revealed that histidine, phenylalanine, tyrosine, and tryptophan biosynthesis and glycine, serine, and threonine metabolism were the top changed pathways. Then, we compared the gene alternations in the metabolomics pathway after retinal detachment using human retina tissues from the GEO public database. These gene alternations and metabolomics changes indicated that decreased histamine, xanthine, and hypoxanthine and increased lipid oxidation were the main physiological processes after RD.

Herein, we reported the main metabolism changes and potential alternations of key pathways verified from the human database. These results provide new information for retinal detachment and may serve as important clues to identify clinical biomarkers.

## 2. Materials and Methods

### 2.1. Animal Ethics and the Animal Model of Retinal Detachment

This study was in accordance with the statement on the use of animals by American Association for Laboratory Animal Science. It was approved by the Institutional Animal Care and Use Committee of Wenzhou Medical University, China (Number: wydw2021-0068).

In all, 25 large male 8- to 10-week-old SD rats weighing about 160 g were obtained from Shanghai Slake Experimental Animal Co., Ltd. (Shanghai, China). All the animals were fed in custom cages at the animal facility separately to prevent contact among them. After they arrived at the animal facility for study, all the animals were fed for 3 days to allow them to adapt to the environment. A total of 12 and 10 animals were randomly assigned to the experiment and control group. The right eye was chosen as the experiment and control eye.

On day 5 after the rats were housed in the new animal facility, retinal detachment was induced in the rats according to the previous process [4]. The right eyes of the 12 animals were selected as the experimental group to be studied. The rats were anesthetized with the drug sodium pentobarbital (i.p., 30 mg·kg^−1^) [4,11]. Then, the pupils were dilated with topical tropicamide (0.5%) and phenylephrine hydrochloride (0.5%) [4,11]. A scleral incision was made 1.5 mm posterior to the corneal limbus and sodium hyaluronate was injected into the subretinal space to ensure that approximately two-thirds of the neurosensory retina detached from the underlying RPE and were floated in the vitreous cavity without complications such as much bleeding et al. The incision was made with a 27-gauge needle to avoid damage to the lens (10 mg·mL^−1^, LG life Sciences, Seoul, Korea). Ofloxacin antibiotic ointment (Dikeluo; Sinqi Pharmaceutical Co., Ltd., Shenyang, China) was applied to the scleral and ocular surface. The rats were monitored daily after surgery.

### 2.2. Sample Collection and Preparation

Three days after establishing the RD model, the rats were anesthetized with the drug sodium pentobarbital (i.p., 30 mg·kg^−1^). Then, the pupils were dilated with topical tropicamide (0.5%) and phenylephrine hydrochloride (0.5%) [4,11]. The eyes were examined under the microscope. Any infectious or bleeding tissues were excluded, and the cornea was cut using scissors, the lens and vitreous were removed, and the remaining retina was separated from the RPE. The retinas were collected, placed on ice, and stored immediately at −80 °C.

Retina samples were homogenized in 800 ul methanol: water (80: 20, *v*:*v*) and 0.5 μL 2-Chloro-L-phenylalanine (1 mg/mL in water) were added for internal standard by using a homogenizer by two cycles of grinding (60 s for each, 40 Hz). Then, 600 μL of supernatant was obtained and dried in a centrifugal vacuum concentrator after centrifugation at 15,000× *g* for 15 min. The dried extract was resuspended with 100 μL of 10% methanol/water (*v*/*v*) for analysis. Quality control samples were prepared by pooling equal aliquots of each sample together, and then they were prepared as other samples previously; every 5 samples followed by one QC sample were injected to monitor the stability [12].

### 2.3. LC–MS/MS Analyses for Untargeted Metabolomics

Retina extracts, QC samples, and solvent blanks were analyzed by Ultimate 3000 UPLC and Orbitrap Fusion Lumos Tribrid mass spectrometer [13]. Metabolites were separated on Waters^TM^ acquity BEH C18 column (2.1 × 100 mm, 1.7 µm) with a flow rate of 0.35 mL/min. H_2_O was used as mobile phase A and methanol was used as mobile phase B in negative ionization mode. While in positive ionization mode, 0.1% formic acid was added into both mobile phases, respectively. The LC gradient was as follows: 0–1 min, 2% B, 1–12.5 min, 2–50% B, 12.5–14.5 min, 50–98% B, 14.5–17.5 min, 98% B; then, mobile phase B returned to 2%. After chromatographic separation, metabolites were ionized b H-ESI. Spray voltage was +3.8 KV/−2.5 KV, and capillary temperature was 320 °C. Full MS scan and data-dependent acquisition was performed to acquire the ms1 m/z and MS/MS information of metabolic features. Full scan (m/z 70–1050) used resolution 120,000 with automatic gain control (AGC) target of 4 × 10^5^ ions and a maximum ion injection time (IT) of 50 ms. MS/MS scan parameters were as follows: resolution 30,000; AGC 5 × 10^4^ ions; maximum IT 54 ms; 1.6 m/z isolation window; Stepped HCD normalized collision energy 15%, 30%, 50%.

### 2.4. Data Extraction and Analysis

Peak areas were extracted from the samples by converting the raw mass spectrometry files to mzXML using ProteoWizard and processed with an in-house program, which was developed using R and based on XCMS (version 3.6.2), for peak detection, extraction, alignment, and integration [13]. Then, an in-house MS2 database (Biotree DB) was applied in metabolite annotation. The cutoff for annotation was set at 0.3. The resultant data were subjected to QC-based normalization (LOESS normalization) and RSD filtering of normalized peaks in the QC samples (RSD < 0.3). The matching of non-targeted metabolite substances is mainly carried out as follows: the molecular weight of the metabolite is determined according to the mass charge ratio (m/z) of the parent ion in the primary mass spectrum, the mass charge ratio of the feature ion generated after fragmentation, and the response intensity of the sub ion. The identification of metabolites and the calculation of the material matching score are based on these previous processes. Finally, to identify the substances, we will analyze the information and compare with standard products in our own and public databases. Our own database combines the information of HMDB, MONA, METLIN, and other public databases.

For pathway enrichment analysis, significant metabolites under NEG or POS mode were combined on the basis of the KEGG human metabolic pathways. Metabolites containing at least two entries were used for analysis. The enrichment ratio was computed by hits/expected. A pathway with *p* < 0.05 was considered significant. The Kyoto Encyclopedia of Genes and Genomes (KEGG) pathway was analyzed to reveal the enriched pathways of the altered metabolites.

### 2.5. RNA-Seq, Sequencing Data Extraction Analysis

The sequencing data (GSE28133) were download from the public database of chips and microarrays in GEO. The data included 38 human retina samples, 19 samples from RD patients, and 19 samples from patients without RD [14]. Gene alternation between detached and intact retinas were compared with |log_2_ Fold Change (FC)| > 1.0 and adjusted *p*-value < 0.05 [15]. After analyzing the alternations of pathways in the main metabolomics, we looked for the key genes corresponding to metabolomics pathway in the public database.

### 2.6. Statistical Analysis

In the untargeted metabolomics, the data were normalized to total peak intensity, and the processed data were uploaded for multivariate data analysis. The data matrix was analyzed using MetaboAnalyst v5.0 and SMICA v16.0 (Umetrics, Umea, Sweden). All the data underwent log10 transformation before analysis. Univariate analysis (ANOVA), partial least squares discriminant analysis (PLS-DA), and orthogonal partial least squares discriminant analysis (OPLS-DA) were performed using MetaboAnalyst v5.0. To evaluate the statistical parameters (accuracy, correlation coefficient (R2), and cross-validation coefficient (Q2)), PLS-DA models were analyzed using the leave-one-out cross-validation (LOOCV) method. To assess the robustness of the model, the 7-fold cross-validation and response permutation testing was carried out. The variable importance in the projection (VIP) for each variable was calculated for its contribution to the classification. VIP > 1 with significant changes was further applied to Student’s t-test at the univariate level, where *p* < 0.01 or *p* < 0.10 was considered as statistically significant. The genes related to metabolomics changes were analyzed between control and retinal detachment groups, and *p* < 0.05 was considered as significant.

## 3. Results

### 3.1. Untargeted Metabolomics of Retina Samples of Rat Model of RD

A microscope was used to evaluate RD, i.e., that the detached retinas were floating in the vitreous cavity after surgery. HE staining was also used to verify successful development of the RD model.

In all, 20 retinal samples (10 RD and 10 controls) were collected for untargeted metabolomics analysis. A total of 1074 peaks were recorded by XCMS records. Principal component analysis (PCA) indicated a good repeatability both in positive and negative models. The QC samples were gathered together closely, which showed good quality control (in Figure 1). The differences in metabolomics between the two groups were identified by OPLS-DA score plots. The results indicated a distinct line between the RD group and the control group, as shown in Figure 1A,B. The permutation analysis of the OPLS-DA model was carried out and proved to be valid and stable in Figure 1C,D.

Significant alternations in metabolites were identified using the criteria of fold change (FC) > 1 and *p* < 0.01. A volcano plot analysis was performed to present the potential changes in metabolites (Figure 2). All the significantly changed metabolites were compound identified within the database, where 60 altered metabolites were identified under the positive group (*p* < 0.01 in 56 of them; Table 1; Figure 3) and 36 altered metabolites were identified under the negative group (*p* < 0.01 in 34 of them; Table 2; Figure 4). The XIC and MS/MS spectra were in the Appendix A.

### 3.2. Involved Pathways Related to Changed Metabolites in RD

The altered metabolites were analyzed, and KEGG was used to discover the involved signaling pathway of metabolites. Different metabolites were obtained in POS and NEG modes (Figure 5). Metabolic pathways are displayed in Figure 5. The top related pathways are (1) histidine metabolism; (2) phenylalanine, tyrosine, and tryptophan biosynthesis; (3) glycine, serine, and threonine metabolism; (4) vitamin B6 metabolism; (5) pyrimidine metabolism; and (6) nitrogen metabolism (*p* < 0.05).

Gene Alternations Verified in the Top-Ranked PATHWAY.

The pathway of metabolomics has been presented previously. The genes in the related metabolism pathways were summarized and compared between detached and intact retinas from human beings in the GEO database.

#### 3.2.1. Histidine Metabolism

Of 15 metabolites, 2 were detected in the histidine metabolism pathway. After RD, histamine and S-adenosylhomocysteine (SAH) decreased and L-histidine increased (Figure 6). L-histamine could be catalyzed by histidine decarboxylase [16]. SAH is the downstream product of L-histidine when methyltransferases catalyze the methyl group in SAM [17]. An analysis of the gene alternations after RD from GEO indicated that histamine N-methyltransferase, histidine decarboxylase, and aldehyde dehydrogenase increased after RD (Figure 6).

#### 3.2.2. Purine and Pyrimidine Metabolism

Of 68 metabolites, 8 were detected in purine metabolism. Xanthine, hypoxanthine, guanine, and guanosine decreased, while deoxyinosine, adenine, and xanthosine increased after retinal detachment (Figure 7A,B). In pyrimidine metabolism after RD, uridine, cytidine, and uracil decreased while thymidine increased (Figure 7A,B). Xanthine and hypoxanthine are catalyzed by xanthine dehydrogenase (XDH) with the byproduct of uric acid in the intracellular spaces [18]. However, the gene alternations of purine and pyrimidine metabolism from GEO indicated that genes of xanthine dehydrogenase, XO, and lactoperoxidase were not changed after RD.

#### 3.2.3. Phenylalanine, Tyrosine, and Tryptophan Biosynthesis

Of four metabolites, two were detected in the phenylalanine, tyrosine, and tryptophan biosynthesis process. After RD, L-tyrosine and L-phenylalanine increased (Figure 7C). Tyrosine is synthesized from phenylalanine by the enzyme of phenylalanine hydroxylase (PHA) [19]. Tyrosine hydroxylase (TH) is the enzyme catalyzing tyrosine to L-3,4-dihydroxyphenylalanine (L-DOPA) [19]. However, the gene alternations after RD indicated that PHA increased (Figure 1D), and TH was not altered after RD. These results indicated that the production of tyrosine from phenylalanine increased after RD.

#### 3.2.4. Glyoxylate and Dicarboxylate Metabolism

Of 16 metabolites, 2 were detected in the glyoxylate pathway. After RD, glycolic acid increased, while citric acid decreased (Figure 7F). Citrate is a substrate for lipid biosynthesis by ATP-citrate lyase enzyme (ACLY) [20]. The gene alternations indicated that ACLY increased after RD (Figure 7E).

## 4. Discussion

In the present study, a total of 90 significant metabolites were found in a rat model of RD. Pathway enrichment analysis by MetaboAnalyst 5.0 indicated that phenylalanine, tyrosine, and tryptophan biosynthesis; histidine metabolism; purine and phenylalanine metabolism; and glycine, serine, and threonine metabolism were profoundly altered after RD.

L-histidine is an essential amino acid and was enzymatically converted to L-histamine, involved in flush sensitivity under dark-adapted conditions [21]. L-histidine was transported to the retina by the L-type amino acid transporter in the retinal capillary endothelial cells [22]. After RD, the blood–retinal barrier (BRB) was disrupted, and the gene of the L-type amino acid transporter increased after RD according to the RNA sequence results (Figure 6). These results indicate that the increase of L-histidine might be because of the disruption of the BRB and the increased transport from capillary endothelial cells by the L-type amino acid transporter.

In vasodilation, histamine is an important neurotransmitter as a regulator of ON ganglion cells and microcirculation [22,23,24,25]. The drosophila visual system recycles the histamine as a neurotransmitter between photoreceptors and other cells [24,25]. Decreased histamine could reduce the threshold of ON ganglion cells to scotopic, full-filed, and flash stimuli [21]. Humans with decreased histamine were found to be less sensitive to light stimuli during the day, and the scotopic b-wave was reasonably reduced [22,23]. S-adenosylhomocysteine (SAH) was the product of S-adenosylmethionine (SAM) when DNA methyltransferases catalyzed the transfer of a methyl group [17]. In our study on analyzing gene alternations in human retina tissues, we found that the metabolite of SAH decreased, and the methyltransferases protein called the isoprenylcysteine carboxyl methyltransferase (ICMT) reduced after RD (Figure 6). ICMT deficiency could lead to photoreceptor dysfunction, progressively diminished rod and cone light-mediated responses, and the defect synthesis of the outer segment in the photoreceptors [26]. Based on the results, we inferred that decreased histamine may influence visual function. The reduced ICMT and SAH might be the reason for the dysfunction of the outer segment of the photoreceptor. These results may explain the outer segment dysfunction and decreased vision acuity in patients with retinal detachment.

Xanthine oxidase (XO) was considered an important factor for reducing oxygen free radicals after retinal injury [27]. XO was mainly detected in capillary endothelium cells and cones in rabbits and played a role in reducing oxygen to toxic intermediates [28]. In our study, we found that XO gene decreases after RD. The amount of xanthine and hypoxanthine decreased after RD (Figure 7B). As XO could reduce oxygen to toxic intermediates, we inferred that the decreased xanthine and XO might lead to higher oxidation after RD. Guanosine was catalyzed by purine nucleoside phosphorylase (PNP) with the product of guanine [29,30]. In the synthesis of guanine nucleotides, inosine-5′-monophosphate dehydrogenase 1 (IMPDH1) was the rate-limiting step [31]. Our study found that the level of guanosine and guanine reduced after RD and therefore could not provide enough sources for the production of inosine monophosphate (IMP). The genes of IMPDH1, guanosine and guanine, reduced after RD, and a previous study has indicated that reduced guanosine and guanine could lead to the dysfunction of photoreceptors [32]. Systemic administration of guanosine could reduce cell death and inflammation after spinal cord injury [33]. Based on these results, we inferred that the decrease of guanosine, guanine, and IMPDH1 might lead to higher oxidation and consequently photoreceptor degeneration.

Tyrosine is a nonessential amino acid and was changed to neurotransmitters such as dopamine, epinephrine, and norepinephrine [34]. It is involved in the activation of various signaling pathways by phosphorylation of the hydroxyl group of proteins called tyrosine kinases, such as VEGF, insulin, and epidermal growth factors [34,35]. On the basis of data from the GEO database, our study also found that tyrosinase-related protein1 (TYR1) increases after RD (Figure 7). In the absence of tyrosine hydroxylase (TH), the neurotransmitters could be produced by tyrosinase [36]. Tyrosinase (TYR) could be expressed by neurons under disease conditions [36]. The tyrosinase-dependent dopaminergic pathway is involved in neurotransmitter and melanin metabolism [37]. Melanin metabolism dysfunction contributes to retinal degeneration [37]. Based on these results, we inferred that the decrease of tyrosine might cause a reduction in the neurotransmitters of retina and consequently lead to photoreceptor degeneration after RD.

Citrate can be used as a substrate for lipid production by enzyme of ATP-citrate lyase (ACLY) [23]. Pyruvate could be catalyzed by pyruvate dehydrogenase (PDH) with the product of AcCoA and lactate by PDH (pyruvate dehydrogenase) [38]. Our gene results indicate that PDH and the rate-limiting glycolytic enzyme PFK1 decreased and ACLY increased, but PDH was not altered (Figure 7). Therefore, we inferred that citrate might be used for lipid production after RD. Based on these results, we inferred that the retina might undergo lipid oxidation rather than lipid biogenesis. Citrate is a key intermediate of the tricarboxylic acid (TCA) cycle in the mitochondria and could be used to for ATP production. Reduced citrate may lead to less ATP production [39]. Lipid oxidation and reduced mitochondrial energy metabolism are the main energy changes after RD.

However, there were different anatomy structures between human and rats’ retinas. The cone density is higher in human fovea than rats, and more cones were detected in the periphery retina in rats [40]. S-cones were densest in the periphery and retinal rim both in rats and humans [41]. The latest study from Robert J. Casson pointed out that short (S)-cones were more susceptible to damage than medium- or long-wavelength cones in the rat model of retinal detachment [11]. This may explain the reason why S-cones are more vulnerable to damage after intense light exposure, diabetic retinopathy, and other human retinal disease. However, the S-cones and rods were different between human and rats. These results may help us to understand why the macular was prone to be damaged in clinical practice. We adopted metabolism to seek the upstream mechanism for photoreceptor degeneration. However, the structure was different between humans and rats. All these results point to a need for more experiments to verify the pathway in retinal detachment, and further to explore the reason for photoreceptor degeneration.

The limitation of the study is that metabolite changes in the animal models were not validated, and the targeted metabolomics were not used to identify the key pathway in retinal detachment. However, we comprehensively analyze the pathway alternations by metabolism and genes from the human retina. These results revealed varying pathological mechanisms involved in retinal detachment, and the neuroprotection of photoreceptors requires multiple approaches. The study helps us to understand the pathology of RD and try to find new targets for neuroprotection.

## 5. Conclusions

In conclusion, our study has revealed the metabolomic profile changes after retinal detachment both in retina tissues of and gene alternations in human retina. Bioinformatics analysis has helped identify histidine metabolism; phenylalanine, tyrosine, and tryptophan biosynthesis; and glycine, serine, and threonine metabolism as the top pathways. All the results indicate varying pathways involved in retinal detachment, such as the decreased histamine metabolomics and xanthine, which may be related to photoreceptor dysfunction, reduced tyrosine metabolomics related to neurotransmitter, and decreased TCA after RD. More research is needed to investigate the key reason and target for neuroprotection.

## Figures and Tables

**Figure 1 metabolites-12-01077-f001:**
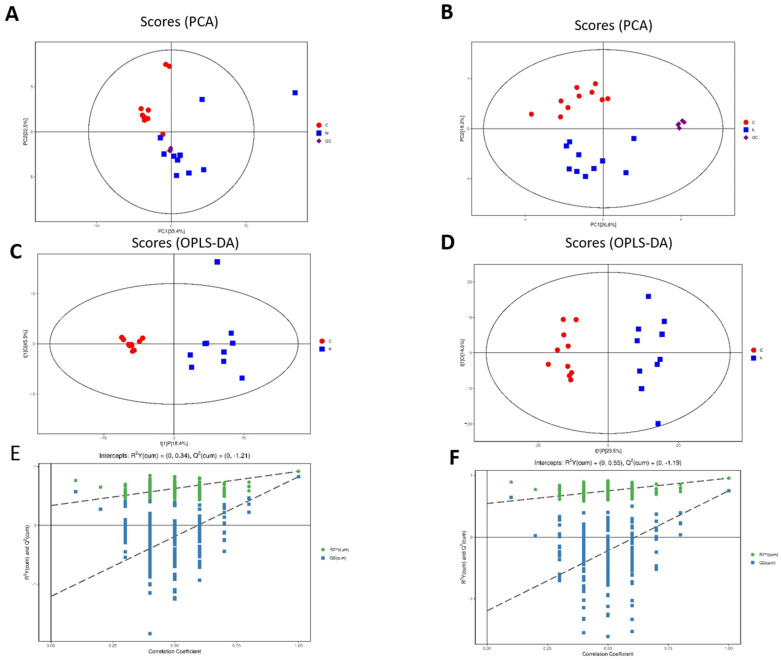
Qualification of untargeted metabolomics analysis. (**A**,**B**) The PCA analysis of the included samples in both positive and negative models. (**C**,**D**) OPLS-DA score plots under the positive and negative models. (**E**,**F**) Permutation analysis plots of the OPLS-DA model under the positive and negative models.

**Figure 2 metabolites-12-01077-f002:**
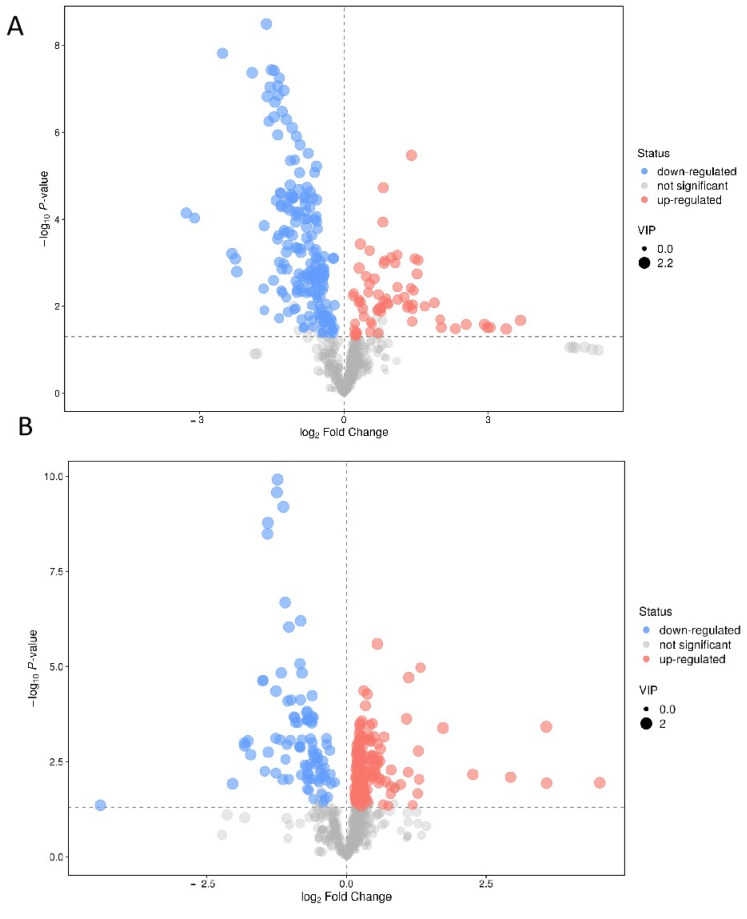
Volcano plots of the untargeted metabolomics under the positive model (**A**) and the negative model (**B**) according to the criteria FC > 1.5 and *p* < 0.05.

**Figure 3 metabolites-12-01077-f003:**
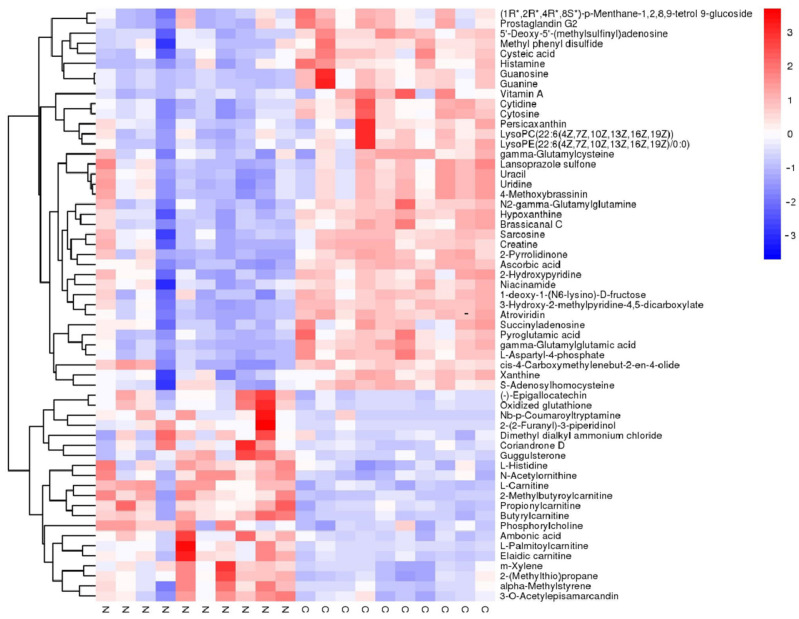
Heatmap of the different metabolites under the positive model in untargeted metabolomics. The blue color indicates the lower relative level of each metabolite, and the red color stands for the higher relative level of each metabolite.

**Figure 4 metabolites-12-01077-f004:**
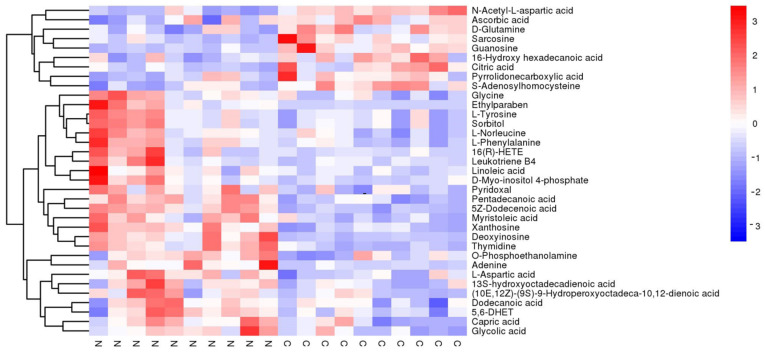
Heatmap of the different metabolites under the negative model in untargeted metabolomics. The blue color indicates the lower relative level of each metabolite, and the red color stands for the higher relative level of each metabolite.

**Figure 5 metabolites-12-01077-f005:**
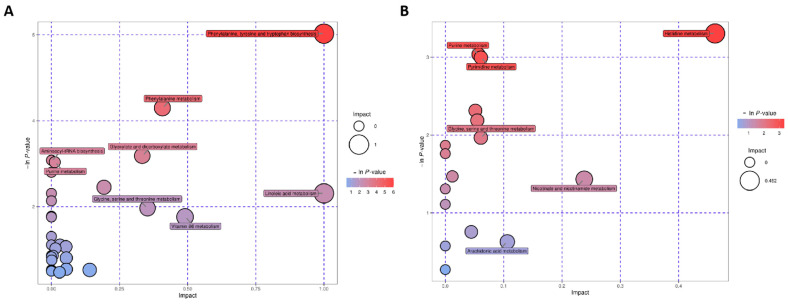
KEGG pathway indicating the top 20 involved pathways of significantly changed metabolites under the positive model (**A**) and the negative model (**B**). The spot size stands for the compound number of metabolites, and the color stands for the *p* value.

**Figure 6 metabolites-12-01077-f006:**
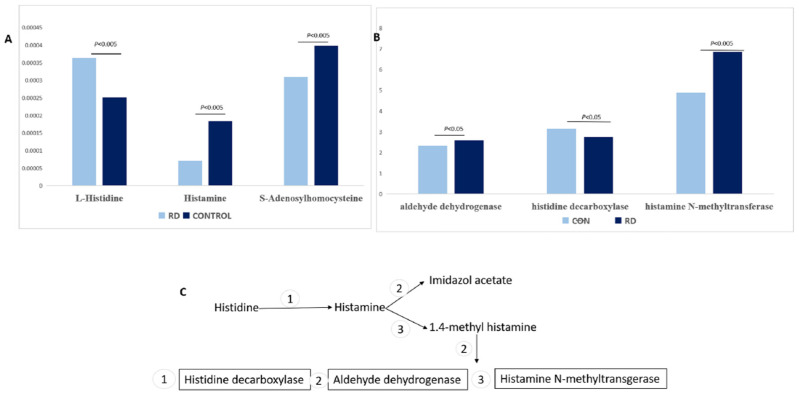
The alternations of histidine metabolism and pathways. (**A**) The alternations in histidine metabolomics by untargeted metabolomics. (**B**) The gene alternations of the histidine metabolomic pathway after RD. (**C**) The summarized pathway of histidine metabolomics, 1 stands for histidine decarboxylase, L-histamine could be catalyzed by histidine decarboxylase, 2 stands for aldehyde dehydrogenase, histamine could be oxidated to imidazoleacetic acid and changed to Imidazol acetate by Aldehyde dehydrogenase, 3 stands for histamine N-methyltransgerase, released histamine is degraded to 1,4-methyl imidazoleacetic acid and it could be changed to 1,4-methylimidazol acetate by Aldehyde dehydrogenase.

**Figure 7 metabolites-12-01077-f007:**
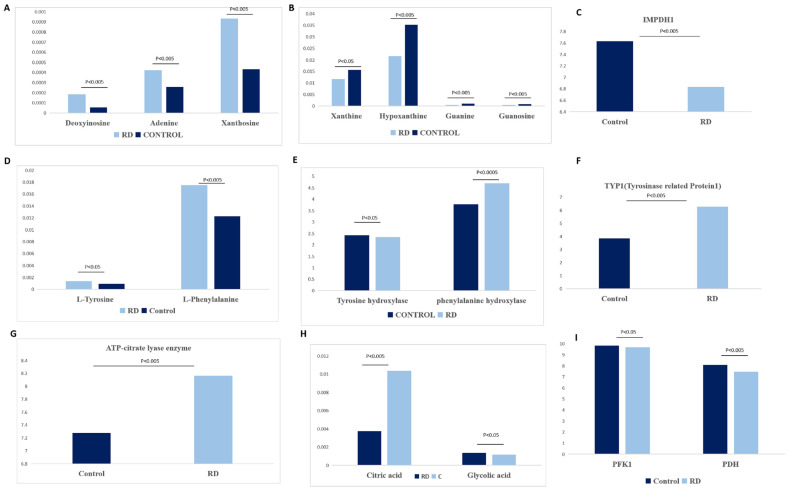
The alternations of main metabolomics and related genes of metabolomics in the pathway. (**A**,**B**) The changes of purine and pyrimidine metabolism. (**C**) Gene change of IMPDH1 after RD. (**D**) The changes of phenylalanine, tyrosine, and tryptophan biosynthesis. (**E**,**F**) The changes of genes in tyrosine hydroxylase, phenylalanine hydroxylase, and tyrosinase-related protein1 after RD. (**G**) Gene change of ATP-citrate lyase enzyme after RD. (**H**) The changes of glycosylate and dicarboxylate metabolites after RD. (**I**) Gene alternations of phosphofructokinase 1 (PFK1) and pyruvate dehydrogenase 1 (PDH1) after RD.

**Table 1 metabolites-12-01077-t001:** Significantly altered metabolites under positive mode by untargeted metabolomics.

Metabolite	VIP	^a^ Fold Change	*p*-Value	mz	rt
Sarcosine	1.588794734	0.753842042	0.00172	90.05489512	43.3375
Cytidine	1.809324897	0.460435551	0.00003	244.0867888	56.1297
2-Pyrrolidinone	2.031331915	0.671816243	0.00004	86.06002251	44.74485
Nb-p-Coumaroyltryptamine	1.905917252	7.582405045	0.02603	307.1416345	496.676
Guanosine	2.153682946	0.198874723	0.00062	284.0989995	154.511
Uridine	1.64987382	0.580089132	0.00488	245.0759403	92.4567
L-Carnitine	1.698905167	1.751054619	0.00012	162.1079781	42.6997
Coriandrone D	1.453741941	1.460356861	0.03866	353.1589536	910.33
Uracil	1.690394112	0.613961856	0.00439	113.0335019	92.8533
2-Hydroxypyridine	1.812738744	0.863836807	0.00079	96.04164293	69.8369
Ascorbic acid	2.04657221	0.57252687	0.00005	177.0330868	51.13885
3-Hydroxy-2-methylpyridine-4,5-dicarboxylate	1.814382779	0.475699949	0.00003	198.0372243	64.1571
L-Histidine	1.529516838	1.44624588	0.00052	156.0754384	39.2151
Niacinamide	1.341138793	0.735454619	0.00428	123.0552253	70.6486
Propionylcarnitine	1.366650349	2.162667902	0.00362	218.1380477	136.899
Pyroglutamic acid	1.825841104	0.403212341	0.00005	130.0498196	82.9935
Cytosine	1.783710152	0.47980553	0.00003	112.050413	57.1205
4-Methoxybrassinin	1.659923965	0.516514641	0.00334	267.0587392	91.8458
Guanine	2.162064502	0.208577348	0.00081	152.0564281	154.536
Hypoxanthine	1.725128074	0.611006413	0.00005	137.0466486	70.23255
m-Xylene	1.921696126	1.243960894	0.00133	107.0826798	939.389
Xanthine	1.457026673	0.743827737	0.00145	153.0433249	79.82625
Phosphorylcholine	1.311159946	1.419536743	0.00481	184.0697118	40.3446
L-Palmitoylcarnitine	1.680174185	4.978528439	0.03278	400.3412165	964.8555
Succinyladenosine	1.480016326	0.618517209	0.00201	384.1150194	330.112
(-)-Epigallocatechin	1.281914517	2.687838842	0.01162	307.0833389	105.983
gamma-Glutamylcysteine	1.16082442	0.633360213	0.02104	251.0704476	68.27515
N2-gamma-Glutamylglutamine	1.79789316	0.502733902	0.00008	276.1224994	44.7747
2-(Methylthio)propane	1.907264524	1.267761325	0.00037	91.05418475	939.711
Dimethyl dialkyl ammonium chloride	1.363211764	1.178980467	0.04779	304.3001743	940.381
S-Adenosylhomocysteine	1.152857028	0.674020148	0.00332	385.1285023	110.14
LysoPC(22:6(4Z, 7Z, 10Z, 13Z, 16Z, 19Z))	1.289484261	0.498294474	0.01125	568.3387229	984.4805
Atroviridin	1.909820405	0.393778382	0.00000	327.0798169	100.4
Oxidized glutathione	1.188118624	4.000915552	0.02011	613.1586921	105.3015
Guggulsterone	1.38032097	1.641203242	0.04156	343.2244804	972.79
(1R, 2R, 4R, 8S)-p-Menthane-1,2,8,9-tetrol 9-glucoside	1.248626769	0.575134248	0.00270	367.1949551	1013.06
Ambonic acid	1.624811991	2.388539493	0.00620	469.3625627	1112.09
Butyrylcarnitine	1.58194961	2.910638976	0.00087	232.154252	279.752
cis-4-Carboxymethylenebut-2-en-4-olide	1.746288625	0.692541355	0.01425	141.0175682	51.75935
Methyl phenyl disulfide	1.260869035	0.681308476	0.00151	157.0143964	51.5829
2-Methylbutyroylcarnitine	1.389115898	2.092890713	0.00103	246.1699963	402.919
Creatine	1.75441836	0.671404514	0.00078	132.076239	43.8692
N-Acetylornithine	1.538467126	1.376210914	0.00205	175.1132487	38.8724
Vitamin A	1.18216123	0.434496736	0.00634	269.2288829	1010.94
1-deoxy-1-(N6-lysino)-D-fructose	1.772390317	0.665660902	0.00050	134.0442436	45.63555
2-(2-Furanyl)-3-piperidinol	2.015285296	10.37822452	0.03335	168.1016873	97.7082
LysoPE(22:6(4Z, 7Z, 10Z, 13Z, 16Z, 19Z)/0:0)	1.68145778	0.420387817	0.00106	526.2975318	985.298
gamma-Glutamylglutamic acid	2.17322362	0.344750109	0.00000	277.1105039	53.96735
5’-Deoxy-5’-(methylsulfinyl)adenosine	1.432320977	0.49083823	0.00000	314.0810526	46.68055
Brassicanal C	1.639967821	0.402276508	0.00003	224.0375107	70.74
alpha-Methylstyrene	1.049964179	1.497412492	0.02551	119.0854447	939.6095
3-O-Acetylepisamarcandin	1.31455005	1.62786731	0.01162	460.2694428	939.137
Elaidic carnitine	1.522562393	3.678436325	0.00832	426.3562901	966.6665
L-Aspartyl-4-phosphate	2.045673216	0.473543833	0.00000	214.0126044	54.0449
Persicaxanthin	1.166835004	0.641843353	0.02979	385.2810119	985.3045
Prostaglandin G2	1.361188168	0.605822078	0.00202	351.2220203	1012.245
Histamine	1.11300757	0.385722377	0.00023	96.92917902	39.3631
Lansoprazole sulfone	1.575610946	0.447892561	0.01350	386.0851118	92.13835
Cysteic acid	1.67467486	0.658755513	0.00010	170.0084242	39.9788

The significant metabolites by univariate analysis under positive mode. Metabolites display a fold change greater than 1.5. ^a^ FC were measured using median values median values between retinal detachment group compared with control group. *p* value *<* 0.05 was considered as significant.

**Table 2 metabolites-12-01077-t002:** Significantly altered metabolites under negative mode by untargeted metabolomics.

MS2 Name	VIP	^a^ Fold Change	*p*-Value	mz	rt
Glycine	1.308701658	1.418244394	0.00497	74.02476574	41.604
Pyrrolidonecarboxylic acid	1.182346811	0.73594283	0.02202	128.0353476	45.9248
L-Norleucine	1.299366092	1.359856849	0.00721	130.0870664	85.1877
Dodecanoic acid	1.013687543	1.112895178	0.02735	199.1699751	998.35
Pentadecanoic acid	1.572438882	1.238311403	0.00004	241.2170986	1032.37
Capric acid	1.22167135	1.160060897	0.00631	171.1355328	957.741
Linoleic acid	1.093421764	1.201474709	0.04900	279.2325558	1036.41
O-Phosphoethanolamine	1.016072298	1.216167453	0.01543	140.0101022	37.523
Adenine	1.446337373	1.629672015	0.01034	134.0472006	133.03
L-Tyrosine	1.272211932	1.527389323	0.00934	180.0663514	71.8062
Pyridoxal	1.319697042	1.421159579	0.00678	166.0464366	108.894
Sarcosine	1.422117232	0.761449281	0.00772	88.04041793	41.2361
L-Phenylalanine	1.429734239	1.42959732	0.00585	164.0714753	191.467
N-Acetyl-L-aspartic acid	1.738817993	0.5753958	0.00001	174.0405148	36.9303
13S-hydroxyoctadecadienoic acid	1.198241881	1.17631196	0.01217	295.2268903	984.563
D-Glutamine	1.213915884	0.729900743	0.00510	145.0688397	39.775
L-Aspartic acid	1.280263363	1.387046842	0.00218	132.0300608	37.6211
Glycolic acid	1.247166563	1.151900036	0.00991	75.00868616	1165.97
Xanthosine	1.754820236	2.161916582	0.00002	283.0677416	231.952
5,6-DHET	1.076984772	1.17425398	0.01023	337.2377561	1000.645
16-Hydroxy hexadecanoic acid	1.047890336	0.864921063	0.01094	271.2201796	982.637
Myristoleic acid	1.390731089	1.52634857	0.00146	225.1854497	1008.66
16(R)-HETE	1.010988944	2.271573423	0.04334	319.2267346	990.646
Ascorbic acid	1.18946295	0.749711824	0.03688	175.0256936	38.9044
Deoxyinosine	1.91372124	3.310838197	0.00041	251.0803053	193.8595
5Z-Dodecenoic acid	1.85788377	1.465643497	0.00000	197.1541993	981.979
Leukotriene B4	1.141316937	1.57686519	0.04224	335.2199433	974.804
Citric acid	1.334618831	0.362946196	0.00560	191.0191466	36.623
D-Myo-inositol 4-phosphate	1.329663611	2.419328351	0.02151	259.0210802	36.9303
S-Adenosylhomocysteine	1.349938422	0.779341895	0.00077	383.1136101	236.586
Thymidine	1.700378459	2.107103442	0.00024	241.0778921	256.103
Sorbitol	1.250229055	1.489605484	0.01152	181.0673795	71.9667
(10E, 12Z)-(9S)-9-Hydroperoxyoctadeca-10,12-dienoic acid	1.020361506	1.198479736	0.02153	311.2223869	966.673
Ethylparaben	1.733042441	23.16190628	0.01128	165.0511808	789.595
Guanosine	1.809686128	0.283277591	0.00120	282.0842135	171.802

The significant metabolites by univariate analysis under negative mode. Metabolites display a fold change greater than 1.5. ^a^ FC were measured using median values median values between retinal detachment group compared with control group. *p* value < 0.05 was considered as significant.

## Data Availability

The RNA sequence data could be analyzed from the website. https://www.ncbi.nlm.nih.gov/geo/query/acc.cgi?acc=GSE28133 (accessed on 29 August 2022).

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
