# Peer review of "Metabolomic Study of a Rat Model of Retinal Detachment"

_metabolites, 2022, doi:10.3390/metabo12111077_

Round 1

Reviewer 1 Report

Dear authors,

I completed the review of the manuscript entitled “Metabolomics Study of a Rat Model of Retinal Detachment”. The manuscript investigates the metabolic changes after retinal detachment using LC-MS approach. Further, it has reported the utilization of transcriptome data from public database (Gene Expression Omnibus) to correlate transcriptome and metabolome changes associated with retinal detachment pathophysiology. Conclusively, the manuscript suggests different metabolic pathway along with lipid oxidation being involved in photoreceptor degeneration.

Below are few of my comments, that would lead to additional review process before recommending for publication.

1)    Proper citation is lacking. For example: GEO, XCMS, Proteowizard etc. The authors should include proper citation wherever required.

2)    Data processing for LC-MS is not clear and authors should clarify about XCMS processing steps. The processing parameters, adduct analysis, isotope correction etc should be mentioned in detail. The description can be included in the supplementary information if space is the limitation.

3)    Importantly, the metabolite identification approach should be mentioned in the manuscript. I would request authors to include how MS1 and MS/MS information was utilized for metabolite ID.

4)    Quality control strategy should be clearly mentioned. I would suggest authors to include the QC preparation method and how the QCs were leveraged in the study design.

5)    Were internal standards used in this study?

6)    Do cationic and anionic models refer to positive and negative ionizations mode in LC-MS data acquisition?

7)    I would strongly recommend authors to include XIC of top 5-10 metabolites along with MS/MS spectra in their supplementary material.

8)    I noticed plenty typos and grammatical errors in the manuscript. As a non-native English speaker/writer, we all have limitation in drafting manuscript. However, I would suggest authors to go for thorough review of the manuscript to correct grammatical errors and typos.

Overall, the content in the manuscript is crucial in understanding mechanism of retinal degeneration. All the comments made should be addressed to proceed further in review process. I would not recommend the manuscript for acceptance until the comments are addressed.

Reviewer 2 Report

The study not only describes the metabolomic composition of the rodent retina in simulated retinal detachment but also aims at identify the key genes corresponding to each metabolic pathway.

Lines 35-37 The authors may consider naming rhegmatogenous retinal detachment as the most common form of detachment. Additionally, do the authors suggest surgical treatment of retinal detachment in age-related macular degeneration in lines 37-38? In cases of submacular hemmorhage?

In line 53 the authors might be interested in mentioning the results of two other important studies on the subject:

1.Metabolomic Analysis of Human Vitreous in Rhegmatogenous Retinal Detachment Associated With Choroidal Detachment.

Yu M, Wu Z, Zhang Z, Huang X, Zhang Q.Invest Ophthalmol Vis Sci. 2015 Aug;56(9):5706-13. doi: 10.1167/iovs.14-16338.PMID: 26313305   2. Metabolomics Analysis of Human Vitreous in Diabetic Retinopathy and Rhegmatogenous Retinal Detachment.

Haines NR, Manoharan N, Olson JL, D'Alessandro A, Reisz JA.J Proteome Res. 2018 Jul 6;17(7):2421-2427. doi: 10.1021/acs.jproteome.8b00169. Epub 2018 Jun 19.PMID: 29877085 Lines 82-92:As the right eyes of 12 animals were the experimental group, did left eyes constitute a control group? Additionally, how did authors check that the sodium hyaluronate was injected into the subretinal space and not sub choroidal space (I mean over RPE and not under RPE)? This is only micrometers difference   Line 97: What do authors mean with "any infectious or bleeding tissues were excluded"? Introduction or Discussion section: Could the authors mention the differences in anatomy and physiology of the rat vs human retina? What could be their impact when comparing the rat retina with human GEO database? Lines 326-327: Is the part of the sentence missing?

Reviewer 3 Report

1.    Avoid the use of abbreviations in the abstract. It would be better to provide a separate list of abbreviations with their full forms.

2.    Although the citation was provided in this manuscript, still the induction of retinal detachment in the rats should be mentioned in brief.

3.    The rats were anesthetized with sodium pentobarbital (i.p., 30 mg.kg-1). Then, the pupils were dilated with topical tropicamide (0.5%) and phenylephrine hydrochloride (0.5%)”. Include at least one to two references for the process.

4.    At line 98: “crystalline and vitreous were removed”, what this was crystalline intended for?

5.    Line 101: 800ul methanol, correct the unit.

6.    Line 102: better to represent it as “2-Chloro-L-phenylalanine”. At line 120: AGC 5×104 ions, correct the unit as 10 raise to the power 4.

7.    Line 104: “centrifugation at 15 000 g for 15 min”, correct the unit for centrifugation.

8.    Put a space between the digits and units, for example “0-1min,2%B,1-12.5min,2%-50%B,12.5-14.5min, 50%-98% B, 14.5-17.5min,98%B; then mobile phase B returned to 2% and so on through the manuscript.

9.    I did not find any reference for LC–MS/MS Analyses for Untargeted Metabolomics, is the method was developed by the researchers for this particular study? If yes, then provide the detail of method development and validation for the same (at least in the supplementary materials).

10. Similarly, in the section 2.5. RNA-seq, sequencing data extraction analysis, not a single reference was cited.

11. The resolution of Figure 6 and Figure 7 is not clear, improve the resolution.

12. The conclusions should be supported with some findings/ data.
